# Anthracnose of Onion (*Allium cepa* L.): A Twister Disease

**DOI:** 10.3390/pathogens11080884

**Published:** 2022-08-05

**Authors:** Ram Dutta, Jayalakshmi K., Sharath M. Nadig, Dalasanuru Chandregowda Manjunathagowda, Vishal S. Gurav, Major Singh

**Affiliations:** 1ICAR-Directorate of Onion and Garlic Research, Pune 410505, MH, India; 2ICAR-Indian Institute of Horticultural Research, Hesaraghatta Lake Post, Bengaluru 530068, KA, India; 3Plant Sciences, Agricultural Scientists Recruitment Board, DARE, Ministry of Agriculture and Farmers Welfare, Government of India, Krishi Anusandhan Bhavan-I, Pusa 110012, ND, India

**Keywords:** anthracnose, *Colletotrichum*, integrated management, onion, twister disease

## Abstract

The onion (*Allium cepa* L.) is a lucrative and high-value vegetable–spice crop in India, but it is sensitive to several of diseases caused by fungi, bacteria, viruses, and nematodes, of which a fungal disease, anthracnose, caused by *Colletotrichum* spp., is a major issue for both onion producers and researchers since it severely affects the bulb production. Twister disease is currently one of the most common problems in onion production, particularly in humid regions, and it reduces productivity while also lowering the value and profitability. Twister disease is visualised by white or pale-yellow water-soaked oval depressed lesions on leaf blades, which are the first symptoms. Lesions expand as the disease advances, and numerous black-coloured, slightly elevated structures/fruiting bodies appear in the middle area, arranged in concentric rings. Curling, twisting, chlorosis of the leaves, and aberrant extension of the neck or pseudo-stem occurs, followed by rotting of the bulb. In an unmanaged crop, an excess gibberellin production by *Colletotrichum gloeosporioides* and *Gibberella moniliformis* is suspected to induce twisting and aberrant neck elongation, which will ruin onion productivity. It is difficult and environmentally unfriendly to control these infections. Since, to the best of our knowledge, this is the first review on onion anthracnose, we tried to consolidate information. This review updates our knowledge of the pathogen, including the disease cycle, infection pathways, and disease management techniques. As a result, growers will be benefit from the application of cultural, biological, and chemical measures and the use of resistant varieties.

## 1. Introduction

The onion, which belongs to the *Alliaceae* family, is one of the world’s most essential and well-known vegetable spices. Onions are consumed as a spice, salad, and vegetable in many Asian countries [1,2]. Central Asia is the primary centre of origin, and the Near-East and the Mediterranean regions are the secondary centres of origin [3]. It is renowned as the “queen of kitchen” in India and is one of the oldest known vegetable crops. It comes in a variety of forms, including fresh, frozen, dehydrated, and green bunching types. Dehydrated bulbs or onion powder are in high demand, as they save money on transportation and storage. It is abundant in proteins and vitamin C and is a good source of minerals (phosphorus and calcium). The onion is noted for its flavour and pungency, and it is utilised as an environmentally beneficial grain protectant [4]. Its therapeutic benefits contribute to its worth. It contains chemical compounds that have anti-inflammatory, anti-cholesterol, and anti-cancer activities [5].

It is grown in a variety of climates around the world, from temperate to semi-arid. Onion is grown worldwide over an area of 3991.51 thousand hectares, with a total production of 76,377.21 thousand MT, of which, fifty percent is grown in Asia [6]. India is one of the world’s largest producer and exporter of onion, and its stands second in the world after China. India is one of the largest producers and exporters of onion globally over an area of 1624 thousand hectares, with a total production of onion estimated to the tune of 266.41 Lakh Tonnes during 2020–2021 (https://nhrdf.org/ (5 June 2022)). Maharashtra alone contributes 40.94 percent of the total area under onion cultivation in the country. Production, on the other hand, has been declining since 1996, resulting in low supply and high prices in the local market, as well as limited quantities of onion for export (NHB 2019). This can be attributed to a number of factors, including a prolonged rainy season and severe pathogen attacks, which have caused the average harvest to drop sharply.

Severe pathogen infections of diseases that drive the average crop to decline dramatically have all been known to contribute to India’s low onion productivity [7]. Among the fungal diseases, anthracnose-twister, purple blotch, downy mildew, and *Stemphylium* blight are the most serious and devastating diseases, reducing both seed and bulb quality as well as quantity [8]. The anthracnose-twister of onion is identified as a rainy season disease was first reported near Zaria, north Nigeria, in 1969 [9], and the projected yield losses at the time were 50–100% [10,11,12,13,14].

In India, it was recorded for the first time in 1981 from the Lonand area of the Satara district in Maharashtra. It was further spread during 1982 and 1983 in Nashik and Pune districts. It was reported from Karnataka in 1987 [15,16,17] for the first time, with severe curls or twister disease of onion in India, which was said to be caused by *Glomerella cingulata*, during *kharif* season. Curling, twisting of leaves, chlorosis, and abnormal elongation of neck portion were major symptoms. *Colletotrichum gloeosporioides* was identified as a causal organism causing anthracnose, necrosis, leaf spot, and fruit rots [18] found in all conditions. 

The leaf infection could result in yield reduction by as much as 100% and rotting of bulb during and after storage for the bulb disease phase. The effective management of onion anthracnose is still lacking. Sound management approaches require a thorough understanding of the infection process, colonisation of the pathogen, and host–pathogen interactions, especially at the sensitive crop stage. However, the initial and post-penetration characterisation of *C. gloeosporioides* has been studied [19,20,21,22,23]. Anthracnose-twister disease of onion caused by *C. gloeosporioides* and *G. moniliformis* remains destructive in most onion fields every cropping season. The disease complex causes 80% to 100% yield loss, which could result in a low supply of onion in the market, resulting in high price [13].

Therefore, quick response and immediate actions are always crucial in addressing this disease because it can affect the entire onion industry as well as leaving a devastating mark on every farmer’s life [13]. It has also been reported that there is still no effective management yet developed against the disease because *G. moniliformis,* which is one of the causal organisms, is a facultative endophyte and hard to control, even by fungicide applications. Seemingly, a single form of management is not enough, and clearly it cannot defeat the two pathogens involved in this disease. However, many studies and experiments were conducted in an attempt to find a management approach that would put the disease at bay. However, we are yet to find proper single solution, and thus integration of different disease management approaches formulated against these two pathogens is needed to generate a series of plans and actions that will result in a strong program against the disease. Moreover, also in onion, biological processes of host–pathogen interactions and ultrastructural studies of *C*. *gloeosporioides* are not clearly understood. In view of the above, the current summing of information in the form of a review will shed light on different aspects of the disease and pathogen(s), which will be useful in terms of covering the grey area in future. To the best of our knowledge, this is the first attempt to consolidate information in the form of review for anthracnose/twister disease of onion in the world.

## 2. Symptomatology

The most prominent symptoms of anthracnose are a small, whitish, water-soaked sunken lesion that appears on leaf sheaths and leaf blades. Later, these lesions become dark oval and elliptical in shape, causing chlorosis, which turns to necrotic regions along the leaf axis. Prominent lesions with concentric spore masses of salmon- or orange-colour waxy acervuli are formed, which later turn to brown to black colour with setae. At the final stages, these lesions become paper-like and brittle; this can easily fall off at the point of lesion. Onion leaf and neck anthracnose symptoms include oval lesions on the leaves and neck, which generally differ in appearance from the commonly occurring onion foliar diseases [24]. In severe conditions, the lesion can be seen on bulb scales, leading to rotting of bulbs and death of the whole plant [9,25,26,27,28].

Twister disease, also known as anthracnose, seven curls, or twister–anthracnose complex, is a leaf twisting disease that has been attributed to *Colletotrichum gloeosporioides* and *Fusarium* spp. [29,30,31,32]; however, the scale of destruction caused by the disease raised questions in terms of whether the two diseases were the same. *C. gloeosporioides* and *Fusarium* spp. cause anthracnose, foliar twisting, curling, and elongation of the elongated neck and slender roots [14,33,34]. In the case of twisting, the leaves are slender with hard tissue, turning yellow to pale green with hindered leaf growth and elongation of the neck region, leading to bending, curling, and twisting of leaves, growing parallel to the soil. Infected bulbs become slender with elongated necks, and the roots become short and sparse, which finally leads to the decaying of bulbs and roots.

### 2.1. Our Initiatives and Research Findings

#### 2.1.1. Anthracnose Symptom Development

There is little known about the time interval for the symptom’s expression in onion anthracnose. Usually, the pathogen cannot be seen under nursery conditions; initial symptoms such as water-soaked lesions appear on the leaf surfaces after 30 days of planting, wherein the lesions turn to white specks and then become chlorotic sunken spots in the middle of the older leaves (Figure 1). Within a week, these spots coalesce and become enlarged over the leaf axis, turning to dark brown and forming larger spots. At later stages, these spots are covered with pathogen propagules, which appear as small salmon- to orange-coloured conidial masses, concentrically later becoming necrotic and developing acervuli/black fruiting bodies all over the necrotised region. The length of the lesions enlarges along with the conidial masses and fruiting bodies along the axis of the leaf. Further, these spread to the fresh leaves, and older leaves fall off. The heavily infected leaf turns completely brown, and leaf blades lead to death, showing dieback of the plant where the lesion develops from the tip of the leaf towards the pseudostem, leaving few leaf blades unaffected. In advanced conditions, the infection moves to the neck region or above the basal plate. Lesions appear on the neck region, forming salmon-coloured conidial masses; concentrically later, the lesions coalesce and enlarge, leading to the death of the entire leaf. In severely affected plants, these lesions superficially appear on onion bulbs, later spreading to inner tissues, where onion bulbs become deformed and lesions develop all over the bulbs on the outer scales. A total of 50% of the bulb is covered with black concentric lesions; following this, the entire bulb is covered with lesions, and finally, complete death of the plant or wilting are observed.

#### 2.1.2. Twister—Anthracnose Complex Development

Initially, a slight twisting of the neck is observed on young plants, and later, abnormal elongation of the neck is seen. As the infection progresses, slight curling and leaf twisting are seen. Later, the neck becomes fully elongated with rolled or curled leaves. In advancement of symptoms, twisted plants become chlorotic, slender, and toppled and also show some white specks on leaves and neck. Then, the lesions become brown and appear in concentric rings. Later, these lesions coalesce and form dark fruiting bodies concentrically. Along with anthracnose twisting, they become severe and form a complex. In severely affected plants, neck and leaves become slender, leading to defoliation or drying of leaves (Figure 2).

#### 2.1.3. Disease Scoring and Assessment

As per our knowledge, there is no proper disease scale for the onion anthracnose and twister complex. As per our visual assessment of the development of the symptoms during crop growth under field conditions, we propose the following rating scale for disease scoring for anthracnose (Table 1) and twister complex (Table 2). With respect to twister disease, there may be a chance of recovering up to 60 percent infection.

## 3. Aetiology

The word ‘Anthracnose’ is derived from the Greek word, which denotes ‘coal’ that produces sunken lesions, and depression in the spot is the unique symptom of anthracnose [35] caused by Deuteromycetes fungi *Colletotrichum* spp. Anthracnose caused by *C. gloeosporioides* in India is a serious problem, resulting in a considerable loss by damaging fruits, vegetables, and medicinally or ethnobotanical useful plants. The genus was recently voted the eighth most important group of plant pathogenic fungi in the world on the basis of perceived scientific and economic importance [36] (Table 3).

In India, the major *Colletotrichum* spp., viz., *C. gloeosporioides, C. falcatum, C. truncatum, C. acutatum*, and *C. coccodes*, are causing anthracnose with a wide host range, including fruits, vegetables, cereals, ornamental plants, grasses, and forest trees [37,38,39,40,41]. In *Allium*, mainly *C. gloeosporioides, C. acutatum*, and *C. coccodes* are causing anthracnose [13,14,33,42], which is suspected to be the cause of twisting and abnormal neck elongation due to excessive accumulation of gibberellins in onions [28,43]. Initially, the possible causes of disease symptoms were suspected as nematodes. However, parasitic nematodes were not associated with infected plants. A fungus named *Fusarium oxysporum* was isolated from infected shallots. Thrips, mites, nematodes, and other fungi were reported to cause similar symptoms [44,45]. Both *C. gloeosporioides* and *F.oxysporum* f. sp. *cepae* were isolated from the infected plants collected at Kalpitiya. It was subsequently reported in common onion cultivation, and a causal organism was identified as *C. gloeosporioides* (Penz). Sacc., the perfect stage of *Glomerella cingulate* Stonem [43,46].

In 2002, the leaf burning of red onion emerged, known as ‘acid disease’, also caused by a fungus, *C. gloeosporioides* [44]. According to the latest report, the causal agent of leaf twister disease of shallot in the Kolonna area has been identified as *C. gloeosporioides* and *F. oxysporum*. *C. gloeosporioides* causes twisting of the above-ground parts, and *F. oxysporum* causes bulb rot. Both fungi were involved in the development of typical leaf twisted disease symptoms observed in the Kolonna area. Similar reports have been observed [47,48,49], and they showed twister disease as complex in nature, being caused by pathogens *Colletotrichum* spp. and *Fusarium* spp.

There are a few reports which have concluded different species of *Colletotrichum* (*C. coccodes, C. truncatum, C. siamense*, and *G. moniliformis*) found to cause twister/anthracnose disease [50]. Some reported that the onion twister menace is due to *C. gloeosporioides, F. oxysporum*, *Sclerotium rolfsii* [51], and *Meloidogyne* spp. [23]. *C. spaethianum* causing anthracnose of *Allium fistulosum* was documented in Brazil [52]. Similarly, *C. spaethianum* infection in *A. ledebourianum* causes anthracnose in India [53]. However, so far, no definite conclusion has been made on the cause of the disease. 

**Table 3 pathogens-11-00884-t003:** Different *Colletototrichum* spp. along with other associated pathogens responsible for causing onion anthracnose.

Pathogen	Host	Reference
*Colletotrichum gloeosporioides*	*Allium cepa*	[9,54,55]
*C. gloeosporioides + Fusarium fujikuroi/Gibberella moniliformis*	*Allium cepa*	[32,56]
*C. gloeosporioides + Fusarium oxysporum*	*Allium cepa*	[43,57,58]
*C. gloeosporioides + F. oxysporum + Meloidogyne graminicola*	*Allium cepa*	[31]
*C. truncatum*	*Allium cepa, A. fistulosum*, and *A. angulosum*	[59,60,61]
*C. dematium f circinans*	*Allium cepa*	[62,63]
*C. spaethianum*	*A. ledebourianum* and *A. fistulosum*	[52,53]
*C. coccodes*	*Allium cepa*	[14,64]
*C. siamense*	*Allium cepa*	[59]
*C. chardonianum*	*Aliium cepa*	[65]
*C. theobromicola*	*Allium fistulosum*	[60]
*C. acutatum*	*Allium cepa*	[13]
*C. scovillei* and *C. nymphaeae*	*Allium cepa*	[66]

### 3.1. Classification

The causal agent of onion anthracnose disease is *Colletotrichum*, which belongs to the Kingdom Fungi, Phylum Ascomycota, Subphylum Pezizomycotina, Class Sordariomycetes, Subclass Hypocreomycetidae, Order Glomerellales, and Family Glomerellaceae [67,68,69]. *C. gloeosporioides* was first identified in 1884, and its perfect state (teleomorph) was given the generic name *G. cingulate* Stonem in 1903. More than 600 synonyms and at least 7 *formae speciales* of the fungus were described, and they are recognised as a heterogeneous group with great variation in morpho-physiological characteristics [70,71]. The fungus produces hyaline; is one-celled; and is an ovoid to oblong, slightly curved, or dumbbell-shaped conidia with obtuse ends. They are usually borne on distinct, well-developed hyaline conidiophores measuring 12.5–14.8 × 4.1–4.7 μm [72]. The waxy acervuli of *C. gloeosporioides* that develop profusely on diseased parts of plants are subepidermal. The acervuli have been reported to measure 80–250 μm [73] and 115–467 × 95–22 μm [53,74,75]. Under moist conditions, the acervuli, when mature, exude pink- or salmon-coloured masses of conidia.

### 3.2. Identification and Characterisation of Colletotrichum spp. and Fusarium spp.

For the effective management of twister disease, accurate identification of *Colletotrichum* species is essential. Classically, identification and characterisation of *Colletotrichum* species have primarily relied upon morphological characters, the optimal growth, the growth rate, the presence or absence of setae, and the existence of the teleomorph *Glomerella* [76]. Conidial morphology has traditionally been emphasised over other taxonomic criteria, although conidia of *Colletotrichum* are potentially variable. Microscopical observation of colony, mycelia, and spore structures helps in morphological identification, but observing the morphological diversity of fungi at the genus and species level is complicated due to the high morphological similarity (Figure 3). Thus, the recent development in molecular biology becomes necessary for generating enormous amounts of genetic data with the aid of molecular markers. The molecular genetic data could be correlated significantly with different physiological and morphological diversities within the same group of fungi [77]. Initially, the molecular characterisation of fungal isolates was performed by the sequencing and blasting of the internal transcribed spacer (ITS) region of rDNA using universal ITS primers [78,79]. However, the markers based on the conserved sequences are insufficient in differentiating between closely related isolates. The development of genic markers for a particular fungal species was found to be mandatory for establishing the correct taxonomic classification. However, these identification techniques are limited in their effectiveness as growth medium, and temperature is known to cause variation in cultural and morphological characteristics, such as size and shape of conidia, colony growth rate, and pigmentation of *Colletotrichum* isolates [80,81]. Thus, the authors of [82] recommended a polyphasic approach for accurate species identification, involving the combined sequence analysis of multiple loci and various morphological data.

Amplified fragment length polymorphism (AFLP) [83] and restriction fragment length polymorphism (RFLP) [84] markers identified 29 *Fusarium* spp. (*F. oxysporum*, *F. proliferatum, F. avenaceum*, and *F. culmorum)* causing disease in onion [83]. The fungus isolated from onion seedlings with twisted and distorted leaves fit the description of *C. gloeosporioides* (Penz.) Penz. and Sacc. [25]—its conidia were aseptate, cylindrical, and hyaline, with 99% sequence similarity to *C. gloeosporioides* accessions after sequencing of the internal transcribed spacer region. All onion *Colletotrichum* isolates tested positive to *C. gloeosporioides* species-specific primer (CgInt2). Arbitrarit rimed PCR analysis using two repeat sequences primers (TCC5 and CAG5) and two microsatellite-derived primers (GACAC3 and MR) demonstrated its potential in species identification and classification of the unknown strains of these two plant pathogens [13].

Similarly, the six isolates were characterised and confirmed as *C. coccodes* as a causal agent for onion twister on the basis of sequence analysis of the internal transcribed (ITS) region of the ribosomal DNA and a 1 kb intron of the glutamine synthase gene (GS) [14]. *C. gloeosporioides, C. acutatum C. fragaria*, and *F.oxysporum* are also associated with the leaf twister infections of red onion [48,57].

ITS regions are known as the barcode locus for fungi [85,86], considered insufficient in delimiting species in the CGSC [87]. While species delimitation using morphology and ITS-based phylogeny remains insufficient for resolution of *Colletotrichum* at the species level, multi-locus phylogenetic analyses have been proven reliable in addressing challenges in the identification of *Colletotrichum* species [88]. In addition to ITS, loci such as glutamine synthetase (GS), glyceraldehyde-3-phosphate dehydrogenase (GAPDH), calmodulin (CAL), actin (ACT), chitin synthase (CHS–1), β-tubulin (TUB2), DNA lyase (APN2), and the intergenic region between DNA lyase and the mating type (Mat1-2) gene (ApMat) have been used to resolve various species in the CGSC [39,52,59].

## 4. Epidemiology

Disease progression is influenced not only by pathogen presence, but also by agronomic management practices and environmental conditions such as temperature, rainfall, and humidity. Environmental factors have a significant impact on the intensity and transmission of any disease or epidemic. The disease spreads because of the favourable host–pathogen and climatic conditions [89]. As a result, before recommending a disease management approach, a complete understanding of the disease’s epidemiology should be addressed. Tropical and sub-tropical regions are the most affected by onion anthracnose. The disease is disseminated more easily in hot and humid environments. Rainfall intensity and duration, humidity, leaf wetness, and light are all crucial environmental elements that influence the severity of the illness. Leaf wetness, for example, has been closely linked to the severity of the disease due to the pathogen’s stronger establishment in term of germination, adhesion, and penetration into host tissues [90]. The association between climatic parameters such as rainfall intensity and duration, prevailing temperature, and humidity, as well as crop geometry and inoculum dispersion, can lead to disease development [91]. Temperature also affects the development of the disease and the presence of leaf wetness, and competitive microbiota further favour the disease development [92]. Temperatures around 23 to 30 °C with a relative humidity of 80% have been reported to be the most optimum conditions for the successful establishment of the disease in a given area [93]. The development of the disease also depends on the host cultivar, along with its resistance to the pathogen.

Onion twister disease spreads from small foci in fields over time. Furthermore, disease incidence was reported year-round, but was particularly high during months of significant rainfall, particularly August–September, when humidity levels were approximately 75–90 percent. The overhead irrigation method utilised in the region may have exacerbated the problem by spreading the inoculum and also by raising the humidity by 85–96 percent, with ideal temperature (20–31 °C) and overcast wet weather in the crop stand. The fungus can spread its spores through soil, water, seed, and agricultural debris [43,57]. Onion anthracnose can easily develop into an epidemic since the pathogen *C. gloeosporioides* completes its cycle in 96 h in onion leaves [42]. This is a classic example of a polycyclic disease whereby the pathogen can easily produce many disease cycles, resulting in millions of spores in one growing season.

### 4.1. Our Initiatives and Research Findings

Although the disease has been studied in several countries during the last 8–9 years, very scanty information is available in India. However, its aetiology and disease cycle is still a mystery, and primary infection is a controversy. To confirm the aetiology and pathogenesis of the disease, infection process studies have been undertaken. In addition, in this review paper, we showcase the disease cycle (Figure 4).

### 4.2. Genetic Diversity and Spatial Distribution of Anthracnose/Anthracnose-Twister Disease

Weather factors that favour the development and spread of the disease are essential to pinpoint the crucial contributions to the development of disease epidemics and are also helpful in the formation of a prediction model or forecasting model [58,94], which have been reported by several countries but not in India. Thus, we initiated survey and surveillance studies in all major onion-growing regions of India in order to study the genetic diversity and hot spots of the disease. Identification and molecular characterisation of the disease and phylogenetic analysis was conducted, identifying the species as *Colletotrichum gloeosporioides*. From our survey, variations were clearly visible in terms of disease distribution and severity. The mean disease severity of anthracnose at Maharashtra, New Delhi, Haryana, and Gujarat ranged between 17.2 and 45.60% (Figure 5) (unpublished data).

### 4.3. Infection Process in Onion by Colletotrichum

*Colletotrichum* has a variety of strategies to infect the host plant, starting with intracellular hemibiotrophic nutrition and progressing to intramural necrotrophic feeding [37]. An intermediate stage showing the partial endophytic lifestyle of the pathogen before adapting to the necrotrophic mode of nutrition in the host plant has been noticed [95]. Different species of this genus exhibit different mechanisms of infection, depending on the host infected.

Direct penetration of spores occurs through intact plant surfaces and through natural openings such as the stomata. At 48 HAI (hours after inoculation), infection hypha germinate from the base of an appressorium and forcibly penetrate the host cell directly through mechanical pressure. Anderson and Walker [19], in a study on watermelon, and Alberto et al. [92], on onion, revealed that *C. gloeosporioides* can exert enough mechanical force to penetrate its host. Direct penetration is the most common means of tissue penetration by necrotrophic fungal pathogens [96]. Moreover, at 48 HAI, an infection hypha from another conidium emerge from the base of the appressorium and penetrate the host through the stomatal opening of the onion leaf. To hasten the direct cuticle penetration, *C. gloeosporioides* also produces cutin-degrading enzymes to dissolve or soften the host cuticle prior to penetration; further on, this was reported in *C. lagenarium* and *C. capsici*. These species can produce esterase and cutinase. An example of this is di-isoprophyl flourophosphate (DFP), which is capable of degrading cutin [96].

Similarly, at 6 HAI in onion, a germ tube emerges from the conidium and a globular-shaped appressorium forms from the endpoint of the germ tube. Between 24 and 48 HAI, the appressoria matures, and an infection hypha emerges through a pore at the base of the appressorium [43]. After 48 h, appressoria penetrates into the host cuticle directly, forming a primary hypha. Simultaneously, the formation of papilla and penetration of the host through stomata also occurs, and there is no evidence of an infection vesicle. The primary hypha starts to branch out, from 48 to 72 h, to form secondary hyphae within the epidermal cells, followed by the massive growth of both intra- and intercellular hyphae, leading to the development of small whitish and water-soaked lesions. At 72 to 96 HAI, intra- and intercellular hyphae radiate from cell to cell, resulting in the formation of acervulli. At 96 HAI, typical onion anthracnose symptoms with salmon-coloured mucilaginous spore matrix can be seen on the infected leaf surfaces. The infection by *F. oxysporum* induces the production of gibberellic acid (GA), leading to abnormal elongation at the leaf tips [49]. In the case of *Colletotrichum* spp., it induces indole acetic acid (IAA) production, leading to abnormal elongation at the neck region. In seed crops, malformation of inflorescence (umbel) leads to poor seed development and also serves as a source of inoculum to the next season. The secondary spread of inoculum is through irrigation water, sprinkler, and wind, accompanied by rain and rain splash.

## 5. Disease Management

Anthracnose-twister disease lowers the bulb and seed output of *Allium* crops by interfering with their production and harvesting. As a result, good management is required to minimise excessive yield losses and maintain high crop output. As part of an integrated illness management strategy, effective approaches for disease management typically combine innate resistance with cultural, mechanical, biological, and chemical factors. Disease losses, as well as chemical and mechanical costs of disease management, may be eliminated by using resistant types. The use of resistant cultivars may be the most cost-efficient, simple, safe, and effective method of disease management.

Anthracnose-twister disease is seed-, soil-, and air-borne and is spread through rain splashes [32]. As it is also known as a disease of the *kharif* season, it should be managed before it spreads throughout the field. Thus, growers should keep an eye on these conditions and use resistant varieties, disease-free seeds, seed treatment with chemicals before sowing, well rotten manure or *Trichoderma*-enriched FYM, and chemicals with some of the strategies to control the disease.

### 5.1. Cultural Management

In cultural control measures, farmers should take necessary actions for onion production as well as to control the disease incidence and spread. These practices should not burden the farmers and should be included in normal agronomical practices. Farmers can take up crop rotation with pulses or cereals for 2–3 years, including using soil sterilisation and disease-free seeds, engaging in timely planting, planting on raised beds, having wide plant spacing, avoiding overhead irrigation, using the recommended dose of fertiliser, and having proper drainage and sanitation, as well as the engaging in the destruction of crop debris, which will reduce the pathogen inoculums. The *kharif* crop planted in flat beds, which is a regular practice by farmers, becomes affected by Anthracnose disease due to water stagnation [97]. Therefore, in order to manage the disease, ICAR-DOGR developed techniques of planting seedlings on a raised bed (BBF), and irrigation through drip or sprinkler needs to be adopted. The late transplanting, use of 15 cm × 15 cm spacing, and a five-day interval of irrigation delays and reduces the occurrence of anthracnose-twister disease development [98].

### 5.2. Botanicals and Biological Management

The usage of fungicides for management of the disease has certain limitations, which include accumulating many toxic compounds in the crop and ecosystem, time-consuming recovery of infected plants, and chemical management being expensive and partially effective. Many researchers have been working on the efficacy of botanicals and bioagents to control the growth of fungal pathogens of onion [99,100,101,102,103,104]; however, very little work is done on anthracnose-twister disease management. Some of the bio-control agents and botanicals were tested against the pathogen both under in vitro as well as in vivo conditions, which showed inhibition against the pathogen growth [105].

Evaluation of different bio-agents and plant extracts against *C. acutatum* and *C. gloeosporioides* in vitro was conducted by [106]. Among the tested bioagents, *Trichoderma harzianum*, *Gliocladium roseum*, *Bacillus subtilis*, *Streptomyces noursei*, and *S. natalensis* inhibited mycelial growth and conidial germination. Similarly, crude culture filtrate extract of *Streptomyces* spp. suppressed the *C. gloeosporioides* growth [107]. The application of *Chaetomium cupreum*, *Penicillium chrysogenum*, *T. harzianum*, *C. globosum*, *T. hamatum*, and a mixture of those bio-products in a powder formulation significantly reduced the incidence of anthracnose disease [108]. Similarly, *T. asperllium*, evaluated for its antagonistic activity against *C. gloeosporioides*, showed 60 percent mycelial growth inhibition [109], and Bacillus subtillis was found to be effective for the management of anthracnose disease [110]. Similarly, eight bio-agents screened under in vitro conditions significantly inhibited mycelial growth of *C. gloeosporioides* over the untreated control. T. hamatum was found to be most effective with the least mycelial growth (16.33 mm) and highest mycelial inhibition (81.11%), followed by *T. virens*, *T. koningii*, *T. harzianum*, *T. viride*, *A. niger*, *P. fluorescens*, and *B. subtilis*. Among a total of 40 isolates of endophytic bacteria isolated from healthy and diseased onion plants, 6 had the greatest ability to inhibit the growth of *Colletotrichum* spp. [111]. A field trial was conducted during *kharif* 2021 to test the efficacy of 13 different Trichoderma spp. against anthracnose-twister disease of onion at ICAR-DOGR, Pune. The results revealed that all the tested Trichoderma spp. performed well in promoting the growth parameters (4 to 8% leaf diameter, 6–27% pseudo-stem girth, and 9–19% number of leaves), 6 to 48% reduction in the disease incidence, and enhancement in bulb yield from 7 to 54% [112]. 

Aqueous extract of *Lantana camara*, *L. viburnoides*, *Echinops* spp., and *Ruta chalepensis* showed effective inhibitory effects on the growth of *C. gloeosporioides* [113]; some of the extracts have been commercialised, and products are available, but they are not used in the disease management on onion [114]. The bulb treatment, together with foliar application of *T. viride*, performed very well in aspects, such as bulb diameter (29.64 mm), circumference of bulb (76.06 mm), mean number of bulbs per bunch (6.95), and yield (130.7 MT/ha), with negligible disease incidence (1.08%) in relation to untreated control [115]. The plant extracts, viz., *Allium sativum, Curcuma longa, Azadirachta indica* leaves, and *Zingiber officinale* were tested against *C. gloeosporioides* in vitro, with it concluded that they have better results in inhibiting the pathogen growth [55].

#### Our Initiatives and Research Findings

We tried using different *Trichoderma* spp., since biocontrol management for onion-anthracnose and twister is very scant. A field trial was conducted during *kharif* 2021 to test the efficacy of 13 different *Trichoderma* spp. against anthracnose-twister disease of onion at ICAR-DOGR, Pune. The results revealed that all the tested *Trichoderma* spp. performed well in promoting the growth parameters (4 to 8% leaf diameter, 6–27% pseudo-stem girth, and 9–19% number of leaves), 6 to 48% reduction in the disease incidence, and enhancement in bulb yield from 7 to 54% [112]. By looking into the all attributes, the treatments *Trichoderma* T-4 R and *T. harzianum* were found to be promising in managing the disease, followed by *Trichoderma*-NRCGT-8 and *T. asperellum*. Combined application of *Trichoderma + Pseudomonas* (1%) of talc formulation as foliar spray was found to be the next best treatment in reducing the twister disease severity. Our investigations found that Trichoderma spp. not only minimises the disease incidence but that it also promotes the biomass of the crop with a high yield. Moreover, the long shelf life of the bulbs produced from Trichoderma-treated plants was also noticed [116]. There are reports on chitin applications for managing anthracnose in other crops. However, it has not been used in managing onion anthracnose [117,118,119]. There are reports wherein chitin stimulates the yield, plant growth, and nutrient content in onion bulb. We have planned to attempt the use of chitin in our program.

### 5.3. Host Plant Resistance

A search for the source of resistance under *Allium cepa* has been undertaken for decades for this disease, but the success has been meagre. A very few germplasms were identified for their resistance that also were moderately resistant against this pathogen. For development of any disease, a favourable environment, inoculum load, crop stage, crop growth, and duration of infection play a pivotal role [120]. Suhardi [121] tested the response of several onion and shallot cultivars to five Indonesian *C. gloeosporioides* isolates. All onion and shallot cultivars were proven to be highly susceptible. Only the shallot cultivar ‘Sumenep’ was partially resistant, which was also observed in field conditions [122]. However, the utilisation of this source of resistance is problematic because ‘Sumenep’ does not flower. A significant variation is found in the survival rate of Latin American onion cultivars to *C. gloeosporioides*. The cultivar ‘Barreiro’ had a survival rate of 40%, whereas ‘Texas Early Grano’ had a survival rate of 4% [123]. Analyses of crosses between both cultivars led the authors to the conclusion that the resistance was polygenic. The resistant sources were tested in wild varieties, which include *Allium galanthum*, *A. roylei*, and *A. fistulosum*, which were found partially resistant [124].

Screening of different onion cultivars against anthracnose occurred, finding IPA 3, Belem, IPA 9, Franciscana IPA 10, Vale Ouro IPA II, and Roxinha de Belem resistant [125]. Similarly, some of the exotic shallot cultivars screened against anthracnose disease led to the finding of three resistant cultivars, viz., Red Grano, Hybrid Rojo, and Primero [48]. Fourteen local and introduced onion varieties were evaluated for their resistance or susceptibility to anthracnose in terms of growth and yield in farmers’ field conditions in Talavera, Nueva Ecija, from January to April 2002. Among the 14 cultivars, only the native cultivar Tanduyong (shallot) was highly resistant to anthracnose [126].

During 2011, 52 genotypes were screened in the natural endemic field condition, with 3 found to be resistant (Ranebennur, RO-282, and Syn 6), 4 moderately resistant (Bhalki, NRCWO 2 (WEC), PKV white, and RO252), and the remainders being highly susceptible with an infection rate of >61% to twister disease [127]. Again, they screened 30 genotypes during 2012 and found 8 to be resistant (Col.744 (dark red), Sel-383-4 (dark red) IIHRDR-12 (dark red) Line-355 (C), Bhima red (C), Sel-12 (red), RO-247 (red), Line-355 (C)), and in their screening, Arka Kalyan was found to be highly susceptible to twister disease. In *kharif*, 15 genotypes were screened against *C. gloeosporioides*; among them, Bhima Raj showed resistance to anthracnose in all the three years [128]. Among 111 onion genotypes, the genotypes W-402 AD-4, DOGR-HY-7, DOGR-HY-56, W-444, MS-100xW197, and W-402 AD-3 showed moderate resistance [129]. A total of 142 were screened against anthracnose; of them, 97 genotypes were disease-free with zero PDI and showed immunity to the disease [130].

#### Our Initiatives and Research Findings

The absence of resistant varieties for anthracnose and anthracnose–twister complex in India or worldwide provided an opportunity to investigate the weaknesses and the response of the onion plants to pathogen invasion. Thus, we undertook high-throughput screening using genotypes including wild *Allium* species under the field in order to find out a promising germplasm which could be incorporated into the population for effective management. A total of 33 genotypes were screened for anthracnose disease; of them, none were found to be resistant to the pathogen. However, four genotypes (accession nos. 1609, 1639, 1655, and 1658) showed a moderately susceptible reaction [131]. In *rabi*, 106 genotypes were screened, and all the accessions were free from anthracnose disease during the season [132].

### 5.4. Chemical Management

Chemicals have been used over time to manage the spread of the fungi, but total control has been a challenge due to the development of resistance by the pathogen. Some of the commonly used fungicides that have been used to control anthracnose disease are mancozeb, carbendazim, propiconazole, bitertanol, hexaconazole, imazalil, and thiabendazole (Table 4).

Chemical management is still the best option to control onion anthracnose-twister disease by evaluating different protectant fungicides and application of growth hormones. The onion plants treated with Captan and carbendazim + paclobutrazol had the highest yield. Protective spray application obviously showed lower disease incidence and severity, shorter neck, and higher yield, as compared to curative spray application. On the other hand, paclobutrazol is a potent regulator of gibberellin biosynthesis and inhibits the oxidation of kaurene to kaurenoic acid [32]. Specifically, it interacts with kaurene oxidase, a cytochrome P-450 oxidase, and inhibits the microsomal oxidation of kaurene, kaurenal, and kaurenol [133]. Reduced levels of gibberellins lead to a decrease in cell division and elongation at the apical meristem of the shoot but have little effect on the production of leaves or root growth [134].

**Table 4 pathogens-11-00884-t004:** Fungicides used against onion anthracnose/twister.

Fungicides with Concentration	Disease	Reference
Benomyl at 0.2%	Onion anthracnose	[135]
Carbendazim and captafol at 10 or 15 g/20 litres	Onion anthracnose	[136]
Thiophanate methyl	Onion twister	[58]
Mancozeb at 0.25%	Onion anthracnose	[127]
Thiophanate methyl 50% + thiram 30% WP and thiophanate methyl 70% wp, and chlorothalonil 70% WP	Onion leaf twister	[45,137,138]
Seed treatment with thiram and spray with zineb 0.25%	Onion twister	[139]
Mancozeb, carbendazim, propiconazole, and thiophanate methyl at 0.1%	Onion twister-anthracnose	[126]
Hexaconazole at 0.1%	Onion twister	[52]
Captan, mancozeb/benomyl Mancozeb or difenoconazole/propiconazole	Onion anthracnose	[140]
Triazoles with gibberellin inhibitor	Onion anthracnose	[141]
Thiophanate methyl	Onion anthracnose	[142]
Trifloxystrobin + tebuconazole, pyraclostrobin + metiram and fluzinam 500 g/L sc.	Onion anthracnose	[143]
Dithane or mancozeb or chlorothalanil or strobilurin fungicides, quadris and cabrio	Onion anthracnose	[144]
Propiconazole at 0.1% and iprobenfos at 0.15%	Onion anthracnose	[145]
Mancozeb 0.25% + tricyclazole 0.1% + hexaconazole 0.1%	Onion twister	[127]
Captan and carbendazim + paclobutrazol	Onion anthracnose-twister	[33]
Carbendazim + mancozeb 0.25%, propiconazole 0.1%	Onion twister	[116]

### 5.5. Integrated Disease Management

In most of the cases, an individual management approach may be ineffective in reducing illness levels to an acceptable level, but the combined effect of numerous strategies may accomplish the objective. Thus, integration of different disease management approaches such as cultural, biological, and chemical forms provides the best management practices to control onion anthracnose. An integrated management of twister disease of onion took place at Kumta, finding that the treatment combination which consisted of soil application of neem cake at 5 q/ha + vermicompost 5 q/ha + *T. harzianum* 2 kg/q of organic matter + seed treatment with *T. harzianum* at 2 g/kg of seeds + dipping the seedlings in *P. fluorescens* solution at 10 g/L + spraying with boron at 2 g/L + hexaconazole at 0.1% + multi K 5.0 g/L at 30 days after sowing resulted in 5.49 percent incidence of twisting and yielded 279.81 q/ha as compared to 41.33 percent incidence and yield of 158.26 q/ha in untreated check [145]. The adoptive module has soil application of FYM (30 t/ha), neem cake 200 kg/ha + seed treatment with vitavax power 2 g/kg + seedling dip with *Pseudomonas* spp. 10 g/L + spraying of (hexaconazole + profenophos) at 0.1%, *Bacillus subtillus at* 10 g/L at 15 days interval, and nutrient module: boron spray (solubor) at 0.2% + Multi K at 0.5%, found to be effective in reducing the twister disease and recording the maximum bulb yield with B/C value of 6.35 [127]. The onion anthracnose management program with benomyl–propineb–difenoconazole–propiconazole (DP)–difenoconazole–carbendazim–mancozeb–*Trichoderma* sp. had the lowest percentage of disease incidence (8.5) and severity (1.2) and obtained good quality and size, the highest number, and the heaviest weight of the marketable bulbs [98].

#### Our Initiatives and Research Findings

We have carried out many investigations to find out effective management practices to control these diseases. Used alone, an individual management practice may not reduce the level of disease to an economic threshold level; hence, the additive effect of several practices are carried out in experiments. Integrated disease management practices are being implemented for seed-to-seed management of the disease. A field trial was conducted during *rabi* 2021 for the management of twister disease. Foliar application of carbendazim 12% + mancozeb 63% 75 WP at 2.5 g/L, *Trichoderma + Pseudomonas* mixture at 10 g/L, and propiconazole 25 EC at 1 mL/L recorded less PDI of 16.59, 17.26, and 18.54, respectively, with high yield of 28.28 t/ha, 27.45 t/ha, and 26.89 t/ha, respectively, and less B/C value (5.74, 5.59, and 5.46, respectively). The lowest yield (14.70 t/ha) was recorded in the control, which had a B/C ratio of 3.03 [116].

During *kharif* 2021, at ICAR-DOGR, India, a field trail conducted with Bhima super was used to evaluate different IDM modules, starting from nursery to 90 days after treatments. It was noted that all the modules inhibited the anthracnose disease, ranging from 7–61% over farmer practice (FP) and 33–51% over existing practice (EP). The maximum (61%) inhibition was recorded with M1 (intensive management), followed by M2 (46%) over FP. When compared with EP, M1 and M2 inhibited 51% and 33% of anthracnose, respectively. Further, M1 also supported 20% and 31% higher yield over FP (18 t/ha) and EP (17 t/ha), respectively, followed by M2 which supported 14% and 24% higher yield over FP and EP, respectively. These modules could be a better option for the management of anthracnose in onion after recommendations [146].

We also implemented different organic formulations for the management of anthracnose, as well as anthracnose-twister disease. Four modified Amritapani-based organic formulations were screened during *rabi* 2020 and *kharif* 2021 for onion crop health, yield, and disease management, with recommended popular variety Bhima shakti (Rabi) and Bhima super (Kharif). The results revealed that DOGROF-3 is the most promising in reducing the disease severity of anthracnose (13.00 PDI) and *Stemphylium* blight (34.00 PDI) compared with the control (61.00 PDI), and also increased the plant growth parameters and yield [147].

## 6. Conclusions and Future Perspective

Onion anthracnose-twister has been studied for decades, but many aspects of the disease’s origin, host–pathogen interaction, and effective therapeutic options remain unknown. There is a pressing need to design an effective integrated management strategy that takes into account the causative agents, epidemiological variables, and pathogenic resistance that contribute to the pathogen’s successful colonisation of host tissues. To create targets for creating resistant onion varieties against the pathogen, genetic diversity and pathogen population information are required. Modifications to the traditional cultural and chemical management must also be considered. On the basis of a few observations, it is recommended that excess moisture levels in onion fields with a high incidence of anthracnose-twister disease be reduced by utilising wider spacing, crop rotation, soil solarisation during nursery, and drip irrigation, which do not generate excess humidity and soil moisture. More research is needed to better understand the infection process, pathogenic variation, biochemical changes, symptom progression, and bulb loss caused by *Colletotrichum* spp. and *Fusarium* spp. during the pre- and post-harvest stages. We still need answers to several questions about this deadly disease complex. The mechanism involved in host–pathogen interaction including physiological and biochemical changes for abnormal elongation of neck and twisting is due to *Fusarium* spp. or *Colletotrichum* spp., which need to be given focus. *Fusarium* spp. and *Colletotrichum* spp. show wide genetic variability, which makes long-term maintenance of resistance in plants difficult. Different mechanisms might play an important role in the emergence of genetic variants, which are yet to be fully understood. These mechanisms, including heterokaryosis, a parasexual cycle, sexual cycle, transposable element activity, and repeat-induced point mutations, could be the points of research for clear understanding of host–pathogen interactions. The identification of effective and stable resistance sources would be useful for resistance to anthracnose-twister disease. The development of genomic SSR and ISSR markers in genotypes needs to be focused on for the identification and isolation of potential disease resistance sources. A comprehensive understanding of the disease triangle will allow for better disease management and improvement in bulb quality as well as quantity. This would develop an effective integrated disease management strategy, limiting the spread of the pathogen and improving host resistance through adopting biological and chemical measures, which ultimately benefits the farming community vis-à-vis the country’s economy. In this regard, we are expanding our research initiatives in all aspects of integrative crop management.

## Figures and Tables

**Figure 1 pathogens-11-00884-f001:**
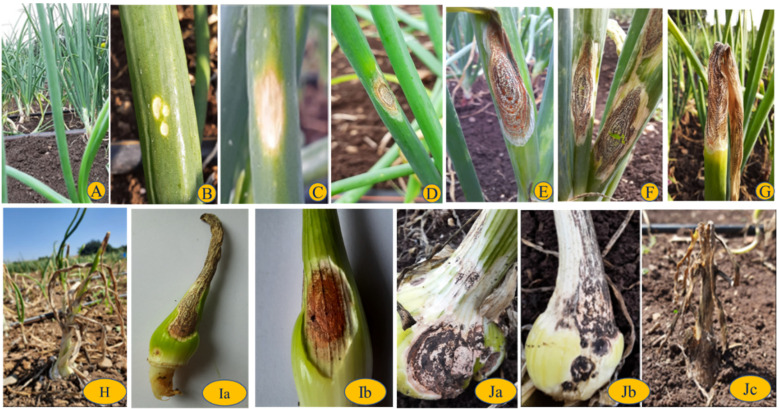
Pictorial representation symptoms of anthracnose disease. (**A**) Healthy; (**B**) grade 1: small white specks; (**C**) grade 2: initial chlorosis; (**D**) grade 3: advancement of lesion; (**E**) grade 4: mature lesion; (**F**) grade 5: lesions begin to coalesce; (**G**) grade 6: advanced lesion leading to death of leaf blades; (**H**) grade 7: most advanced stage lesions, wherein the dieback appearance of the plant occurs, leaving few leaf blades unaffected; (**Ia**) grade 8: advanced lesions on neck region of the plant; (**Ib**) grade 8: salmon-coloured conidial mucilage on lesion on the neck region; (**Ja**) grade 9: 50% of the bulb covered with lesion; (**Jb**) lesions coalesce and form black fruiting bodies on the entire bulb; (**Jc**) complete infection, the death of the plant.

**Figure 2 pathogens-11-00884-f002:**
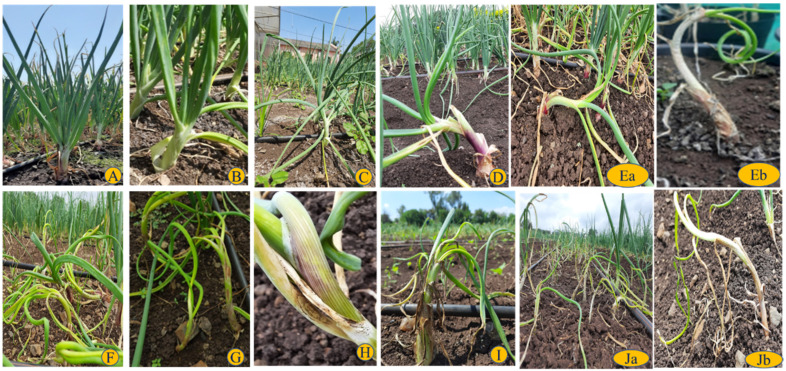
Twister–anthracnose complex: (**A**) healthy; (**B**) grade 1: slight twisting from the neck; (**C**) grade 2: slight elongation of neck and twisting; (**D**) grade 3: twisting of the leaves with neck elongation; (**Ea**) grade 4: twisting with severe neck elongation; (**Eb**) grade 4: elongated neck with leaf curling; (**F**) grade 5: severe stage of twisting falling of plants on the ground and neck, and foliage becomes slender; (**G**) grade 6: leaf twisting and initial anthracnose lesions; (**H**) grade 7: lesion advancement with fruiting body formation; (**I**) grade 8: twister anthracnose complex leading to death of old and young leaves; (**Ja**,**Jb**) grade 9: twister anthracnose complex leading severe neck and foliage drying and defoliation or wilting.

**Figure 3 pathogens-11-00884-f003:**
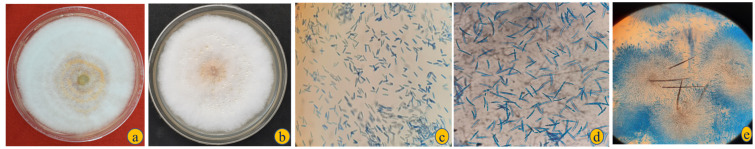
Colony morphology of (**a**) *Colletotrichum gloeosporioides* and (**b**) *Fusarium* spp.; conidial morphology of (**c**) *Colletotrichum gloeosporioides* (400×), (**d**) *Fusarium* spp. (400×), and (**e**) Acervulli.

**Figure 4 pathogens-11-00884-f004:**
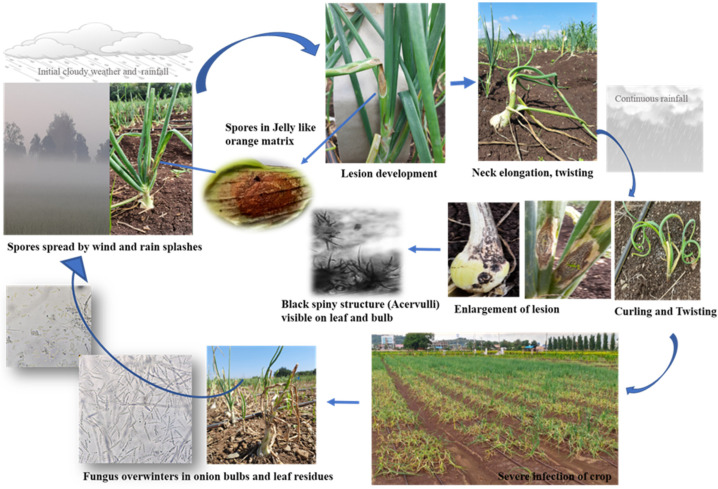
Disease cycle of anthracnose/twister–anthracnose complex in onion.

**Figure 5 pathogens-11-00884-f005:**
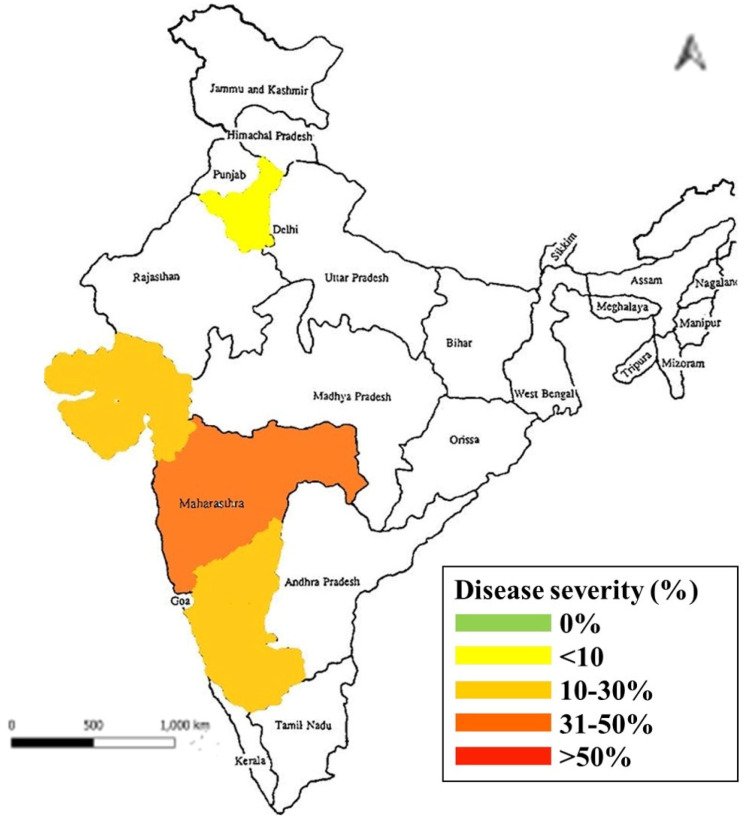
Map showing the disease severity of anthracnose disease in India.

**Table 1 pathogens-11-00884-t001:** Disease rating scale for anthracnose development assessment.

Rating Scale	Description	Corresponding % Damage	Pictorial Representation as per Figure 1
1	Small white specks	0.1–1	B
2	Chlorotic spots	1.1–2	C
3	Advancement of the lesion concentric rings	2.1–6	D
4	A mature lesion with salmon/orange-colored conidial mass	3.1–11	E
5	Lesions began to coalesce	11.1–21	F
6	Advanced lesion leading to death of leaf blades	21.1–31	G
7	Most advanced stage lesions; the dieback appearance of the plant leaving few leaf blades unaffected	31.1–41	H
8	Advanced lesions on the neck region	41.1–61	Ia,b
9	Complete infection; the death of the plant/black fruiting bodies on the entire bulb	61.1–100	Ja–c

**Table 2 pathogens-11-00884-t002:** Disease rating scale for twister–anthracnose complex assessment.

Rating Scale	Description	Corresponding % Damage	Pictorial Representation as per Figure 2
1	Slight twisting from the neck	<10	B
2	Slight abnormal elongation of the neck and twisting	11–20	C
3	Twisting of the leaves with abnormal neck elongation	21–30	D
4	Twisting with severe neck elongation with leaf curling	31–40	Ea,b
5	Severe stage of twisting falling of plants on the ground and neck and foliage becoming slender	41–50	F
6	Leaf twisting and initial anthracnose lesions	51–60	G
7	Lesion advancement with fruiting body formation along with twisting	61–70	H
8	Twister anthracnose complex leading to death of old and young leaves	71–80	I
9	Twister anthracnose complex leading to severe neck and foliage drying and defoliation or wilting	>81	Ja,b

## Data Availability

Not applicable.

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
