# Peer review of "Anthracnose of Onion (Allium cepa L.): A Twister Disease"

_pathogens, 2022, doi:10.3390/pathogens11080884_

Round 1

Reviewer 1 Report

The Review article “Anthracnose of onion (Allium cepa L.): A twister disease” is it interesting but not new topic in the literature. However it is a good review article.

A little mistakes:

[90, 94, 133, 141] no literature in the text

Line 28 – Alliaceae – italic

Line 45 – Onion – with a lowercase letter

Line 72 - without parentheses behind [23-27]

Line 78 – Also - with a capital letter, dot after [28]

Line 102 - without parentheses behind [29]

Line 108, 203 – dot after spp

Line 184 – it should be [48-49]

Line 199 not Italic [53]

Line 216 without parenthesis before [74]

Line 298 without parentheses behind 47

Line 256 – gloeosporioides - with a capital letter

Line 315 – Survey - with a lowercase letter

Line 319 – after severity should be dot

Line 320 – should be Figure 5 not Figure 3-4

Line 364, 486 – Allium – italic

Line 401-422, 432-445, 539, 574-579 – name of microorganisms – italic

Line 402, 411 – in vitro - italic

Line 479 without parenthesis before [127]

Line 522 without parentheses behind [22]

Line 518-526 which means - @- in the text?

Reviewer 2 Report

It is interesting work, although it is not exactly a review paper. I would say that it states preliminary results of research work by combing some review information as well. However, it is interesting work and I would suggest its publication after thorough revision by the authors. 

General comments

-        -   It is good that you provide such detailed information on the disease status about India, but it would be nice to add information about the disease status at other countries as well (e.g. Mediterranean countries). It is a review paper and as such the interest would be increased for the readers.

-         - Please use italics when you refer to fungi names in the text

Specific comments

Lines 307-308: why would the disease cycle may differ in India than the disease cycle of the pathogen in other countries. Please explain.

Line 314: Again… please report what is known about the forecasting models in other countries and why the same models cannot be used in India.

Line 318: Please indicate the results of the research. There is no point to state that the research was undertaken. The results are important for the readers.

Lines 379-391: what about alternate crops? Or trap crops?

Line 401-419: please convert to italics the fungus names

Lines 423-430: Has any of those extracts been developed to a commercial product?

Lines 431-445: please convert to italics the fungus names

Lines 441-445: what about chitin applications? Any research on application of chitin? Chitin applications are very effective against other soil pathogens and I believe that has the potential to be used against anthracnose. Moreover, Trichoderma spp are growing very well with chitin and chitin helps in the establishment of Trichoderma populations in the soil.

Lines 570-581: please convert to italics the fungus names

-      
